# Tri-Reagent Homogenate Is a Suitable Starting Material for UHPLC-MS Lipidomic Analysis

Olatz Fresnedo [1], Beatriz Abad-Garcia [2] and Yuri Rueda [1,*]

1 Lipids & Liver Research Group, Department of Physiology, Faculty of Medicine and Nursing, University of the Basque Country (UPV/EHU), 48940 Leioa, Spain
2 Central Analysis Service (SGIker), Martina Casiano Platform, University of the Basque Country (UPV/EHU), 48940 Leioa, Spain
* Correspondence: yuri.rueda@ehu.eus

**Abstract:** Background: Transcriptomic and lipidomic dual analyses usually initiate with independent extractive procedures. That entails a difficulty in aligning results from both omics platforms, especially in the case of highly heterogeneous tissues, such as the kidney. Methods: Bligh and Dyer lipid extraction was performed using rat kidney homogenates prepared in PBS or commercially available Tri-reagent used for RNA extraction. Samples were analyzed by ultrahigh performance liquid chromatography-mass spectrometry (UHPLC-MS) lipidomic analysis. Results: Comparison of the lipidome obtained from phosphate-buffered saline (PBS) and Tri-reagent homogenates showed qualitative and quantitative validity of the Tri-reagent homogenate with the exception of ether lipids; the acidic nature of the mix seems to promote the hydrolysis of the ether bond, especially in plasmalogens. We tested several conditions in the sample processing, which allowed to optimize the procedure. Conclusions: Aiming to implement a method that allows the extraction of RNA and lipids from the same tissue homogenate not using external tracers, we here report the use of Tri-reagent homogenates as a suitable starting material for UHPLC-MS lipidomic analysis.

**Keywords:** lipidomic; kidney; lipid extraction; RNA extraction; Tri-reagent; UHPLC-MS





## 1. Introduction

Transcriptomic and metabolomic dual analysis in biological systems enables to integrate the primary gene expression with phenotypic responses. Classic methodologies initiate with independent extractive procedures in each omics workflow. Regarding RNA extraction, the use of commercially available Tri-reagents based on the well-established acid guanidinium thiocyanate-phenol-chloroform extraction method [1] is one of the preferred protocols. A careful handling of samples usually renders very good quantitative and qualitative recoveries of total RNA. The procedure includes a step of phase separation using chloroform. As described by Podechard et al. [2], the discarded hydrophobic phase can be used as a source of material for lipid extraction in lipidomic analyses by gas chromatography. To assess the representativeness of the chloroformic phase, they used a radioactive fatty acid as a standard [2].

Adding an external radioactive label to samples adds uncertainty to the whole protocol and restricts its use to suitable facilities. To avoid this factor, in this work, we tested the use of tissue homogenized in Tri-reagent, usually used for RNA extraction, in a lipidomic analysis by ultrahigh performance liquid chromatography-mass spectrometry (UHPLC-MS). Comparison of the results with a usual lipidomic analysis starting from tissue homogenized in a standard buffered solution shows that Tri-reagent homogenate is a suitable starting material for UHPLC-MS lipidomic analyses. Although the acidic nature of Tri-reagent induces the hydrolysis of some ether-containing lipids, we describe that keeping the time between tissue homogenization and lipid extraction short and the reinforced buffering of the extraction mixture solves the problem. A common extractive procedure will be useful

for multi-omics approaches in studies of heterogeneous tissues/organs, such as the kidney, whose complex histological structure is reflected in differential lipidomes as it has been revealed by imaging mass spectrometry of human and mouse samples [3,4]. Investigations on other tissues with heterogeneous lipidome distribution, such as brain [5] or cancerous tissues, which are characterized by intra-tumor heterogeneity [6–8], would benefit from the protocol described here.

## 2. Materials and Methods

### 2.1. Preparation of Lipid Extracts

To prepare lipid extracts, kidneys were obtained from three female Sprage-Dawley rats. Animal handling procedures were performed according to the University of the Basque Country ethical committee (M20/2016/237). Samples were maintained in solid $CO_2$ until homogenization and kept at 0 °C during homogenization. A kidney from each animal was finely minced using a scalpel, and all tissue fragments were pooled. Aliquots of the tissue pool were homogenized in 10 volumes of phosphate-buffered saline (PBS; 10 mM phosphate buffer, pH 7.4, 150 mM NaCl) or Trizol, the Tri-reagent from Invitrogen (Waltham, MA, USA), using a Polytron homogenizer (Kinematica AG, Malters, Switzerland) (12 mm dispersing aggregate, 1 min at 80% of maximum intensity).

Protein concentration was measured using the BCA assay (Thermo Scientific, Waltham, MA, USA). As the Trizol reagent is incompatible with the BCA assay, we used the protein concentration measured in PBS-homogenate as a reference. Protein contributes to around 10% of fresh kidney mass.

In the first experiments, homogenate volumes containing 0.1, 0.3 and 0.5 mg of protein of PBS homogenates and the equivalent tissue masses from Trizol homogenates were used for lipid extraction. Homogenate volumes were adjusted to 600 µL with PBS prior to lipid extraction.

In the last experiment, the same procedure was followed (only with 0.5 mg of protein), but in some cases, 3× concentrated PBS was used to increase the buffering of the extraction mixture. In addition, in order to observe the effect of time between homogenization and addition of homogenate to the lipid extraction mixture, time was controlled and kept below 2 min or over 5 min.

Lipid extraction is based on the procedure described by Bligh and Dyer [9]. Briefly, 600 µL of the sample were added to 9 mL of chloroform:methanol 1:2 (*v:v*). At this step, internal lipid standards were added to allow quantification in the lipidomic analysis: Splash LipidoMix, Ceramide/Sphingoid Internal Standard Mixture I, Cardiolipin Internal Standard Mixture, D18:1/12:0 monosulfogalactosyl (β) ceramide, 24:0 (d4) L-carnitine and oleic acid (d9), all from Avanti Polar Lipids (Alabaster, AL, USA). The following steps were performed at room temperature: vortex mix (2 min), add 3 mL of chloroform, vortex mix (1 min), add 4.8 mL of water, vortex mix (1 min). Samples were centrifuged (1200× *g*, 15 min, 4 °C) for phase separation. The lower, chloroformic phase was collected, and the upper, aqueous phase was re-extracted at room temperature: add 7.2 mL of chloroform:methanol:water 1:1:1 (*v:v:v*), vortex mix (2 min). Samples were centrifuged as done previously, the chloroformic phase was collected and combined with the previous one. Chloroform was evaporated by centrifugal evaporation (Savant SpeedVac, Thermo Electron Company, Waltham, MA, USA) and samples were stored at −80 °C in a $N_2$ atmosphere until the lipidomic analysis. Chloroform and methanol were purchased from Scharlau (Sentmenat, Spain) and were of ≥99.8% purity.

### 2.2. UHPLC-MS Analysis

Lipidomic analysis was performed in the Central Analysis Service facility of the University of the Basque Country (SGIker UPV-EHU, Campus of Biscay, Leioa, Spain), and the procedure was published previously [10]. Global lipidomic profiles were determined by tandem MS using an electrospray ionization source (ESI) in negative (−) and positive mode (+) after separation of lipid classes by a reverse-phase ultrahigh performance liquid

chromatography (UHPLC). The chromatographic separation was achieved on a Vanquish UHPLC system (ThermoFisher Scientific, Waltham, MA, USA), equipped with a binary solvent delivery pump, a thermostated autosampler and a column oven. A reverse-phase column (Acquity UPLC C18 CSHTM 2.1 × 100 mm, 1.7 μm) and a pre-column (Acquity UPLC C18 CSHTM 2.1 mm × 5 mm, 1.7 μm: VanGuard, Valley Forge, PA, USA), both purchased from Waters (Milford, MA, USA), were used at 65 °C to separate individual lipids. The mobile phases consisted of acetonitrile and water (40:60, *v:v*) with 10 mM ammonium formate and 0.1% formic acid (phase A), and acetonitrile and isopropanol (10:90, *v:v*) with 10 mM ammonium formate and 0.1% formic acid (phase B). The applied elution conditions were: 0–2 min, 40–43% B; 2–2.1 min, 43–50% B; 2.1–12 min, 50–54% B, 12–12.1 min, 54–70% B; 12.1–18 min, 70–100% B. Finally, washing and reconditioning of the column were done. The flow rate was 500 μL/min, and the injection volume was 2 μL. All samples were kept at 10 °C during the analysis. Optima$^{®}$ LC/MS-grade water, methanol, acetonitrile, isopropanol and formic acid were obtained from Fisher Scientific (Waltham, MA, USA). Ammonium formate was purchased from Sigma-Aldrich (St. Louis, MO, USA).

All UHPLC-MS/MS data were acquired on a Q Exactive HF-X hybrid quadrupole-Orbitrap mass spectrometer (ThermoFisher Scientific, Waltham, MA, USA) equipped with a HESI (heated electrospray ionization) source using a data-dependend LC-MS/MS method (top 15 MS2) in both positive mode and negative mode. The mass spectrometer settings were optimized using the Splash LipidoMix and Ceramide/Sphingoid Internal Standard Mixture I (Avanti Polar Lipids, Alabaster, AL, USA). The flow rates of sheath gas, sweep gas and auxiliary gas for both polarities were adjusted to 35, 0 and 10 (arbitrary units). For both ionization modes, the capillary temperature and the heater temperature were maintained at 285 °C and 370 °C, respectively, while the spray voltage was 3.90 kV for positive and 3.20 kV for negative ionization. The S-lens RF level was set at 40. The Orbitrap mass spectrometer was operated at a resolving power of 120,000 in full-scan mode (scan range 250–2000 *m/z*, automatic gain control target $1 \times 10^6$) and 7500 in Top15 data-dependent MS2 mode (HCD fragmentation with a stepped normalized collision energy of 25 and 30 in positive mode, and 20, 30 and 40 in negative ion mode; injection time 11 ms; isolation window 1 *m/z*; automatic gain control target $1 \times 10^5$ with a dynamic exclusion setting of 6.0 s). The spectrometer was calibrated externally every three days within a mass accuracy of 1 ppm.

*2.3. MS Data Processing*

All the MS data were acquired and processed using the Xcalibur software package (version 4.1, Thermo Fisher Scientific, Waltham, MA, USA), while the LipidSearch software version 4.2.27 (Mitsui Knowledge Industry, Tokyo, Japan) was used to identify and quantify the lipid species in these complex biological samples. The key processing parameters were as follows: target database, General; precursor tolerance, 5 ppm; product tolerance, 5 ppm; product ion threshold, 1%; m-score threshold, 2; Quan *m/z* tolerance, ±5 ppm; Quan RT (retention time) range, ±0.5 min; use of main isomer filters and ID quality filters A, B, C and D; adduct ions H+, Na+ and NH4+ for positive ion mode, and H− and HCOO− for negative ion mode.

A variation coefficient threshold of 30% was applied to the intensities of masses assigned to lipid structures from 10 runs of the quality control mix to filter the data acquired. Table 1 summarizes the lipid classes detected and the analyzed adducts (those of highest intensities in each lipid class).

**Table 1.** Lipid classes detected by UHPLC-MS after lipid extraction from PBS and Trizol homogenates of kidney. Abbreviations (Abbr.) and the adduct of the highest intensity are indicated for each lipid class.

| Lipid Category | Lipid Class | Abbr. | Most Intense Adduct |
|---|---|---|---|
| Glycerophospholipid | Phosphatidylcholine | PC | [PC+H]+ |
| | Ether-PC | PCe | [PC(O)+H]+ |
| | Lyso-PC | LPC | [LPC+HCOO]− |
| | Phosphatidylethanolamine | PE | [PE-H]− |
| | Ether-PE | PEe | [PE(O/P)-H]− |
| | Lyso-PE | LPE | [LPE-H]− |
| | Phosphatidylserine | PS | [PS-H]− |
| | Phosphatidylinositol | PI | [PI-H]− |
| | Phosphatidylglycerol | PG | [PG-H]− |
| | Cardiolipin | CL | [CL-H]− |
| Sphingolipid | Sphingomyelin | SM | [SM+H]− |
| | Hexosylceramide | HexCer | [HexCer+Na/H]+ |
| | Sulfatide | ST | [ST+HCOO]− |
| | Ceramide | Cer | [Cer+HCOO]− |
| Neutral glycerolipid | Triglyceride | TG | [TG+NH$_4$]+ |
| | Diglyceride | DG | [DG+Na]+ |

## 3. Results and Discussion

This work aims to explore whether Tri-reagent homogenates are appropriate for lipid extraction. To do that, we used Trizol, Tri-reagent manufactured by Invitrogen. In our first attempt to prepare lipid extracts, we used the chloroformic phase from the Trizol reagent protocol for RNA extraction, as described previously [2]. To avoid the use of radioactive tracers, we recovered the hydrophobic phase (usually discarded) in a standard RNA extraction (50–100 mg tissue in 10 volumes of reagent). After extraction of lipids by the Bligh and Dyer method and evaporation of chloroform, a non-volatile residue remained in the extracts that made it unsuitable for subsequent UHPLC-MS analysis. That is why we explored the possibility of using the whole Trizol homogenate to extract lipids. This allowed working with a homogeneous suspension and a representative aliquot for lipid extraction.

To establish the validity of Trizol homogenate as starting material for the determination of lipid composition of kidney, we performed a lipidomic analysis. Representative base peak intensity chromatograms obtained in ESI-positive and -negative modes of the UHPLC-MS analysis are shown in Supplementary Figure S1. From the data of the first experiment, we calculated the average percentage that each lipid species' intensity represents in its lipid class and compared the result with that from PBS-homogenate (Supplementary Figure S2). Only classes with more than five molecular species identified were considered for this analysis. The net values detected in the extracts are listed in Supplementary Table S1. With the exception of lysoPE species, the lipid composition is almost identical in both extracts. Among lysoPEs, LPE(20:4) is the most abundant species in the Trizol extract, while LPE(16:0) and LPE(18:0) are much lower than in PBS extracts.

Next, to evaluate the quantitative response of the lipidomic analysis, we performed a linearity analysis in lipid extracts from 0.1, 0.3 and 0.5 mg of protein. The objective was dual; on one hand, we wanted to check whether lipid detection from Trizol homogenates showed a good linear response against protein quantity. On the other hand, we wanted to rule out a possible artifact in LPE quantification. Lipidomic analysis results are shown as the sum of intensities of lipid species of each lipid class (Figure 1); again, only classes with more than five molecular species identified were considered for this analysis. Values are plotted against the protein quantity of kidney mass used for lipid extraction; R$^2$ values for each linear regression analysis are shown.

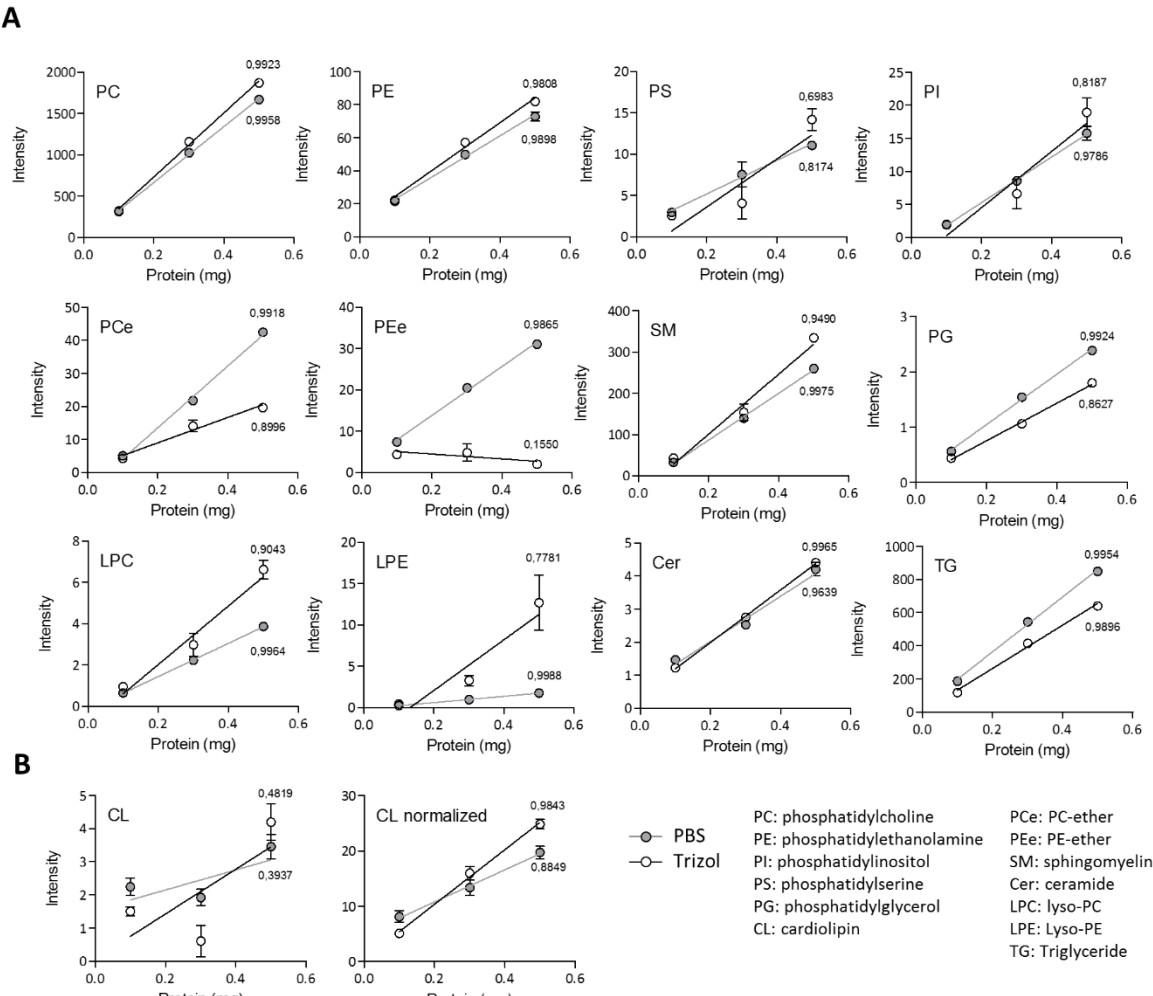

**Figure 1.** Lipid class intensity detected by UHPLC-MS after lipid extraction from Trizol or PBS homogenates of kidney. (**A**) Values represent the sum of intensities of lipid species of each lipid class; (**B**) values corresponding to CL are represented as in (**A**) and normalized with intensity of the internal standard included before lipid extraction (Cardiolipin Mix I from Avanti Polar Lipids). Values are plotted against the protein quantity of kidney homogenates used for lipid extraction. $R^2$ values for each linear regression analysis are shown next to each line. Only classes with more than 5 molecular species identified have been considered for this analysis. 2 (0.1 mg of protein) or 3 (0.3 and 0.5 mg of protein) homogenates were used for lipid extraction and data are shown as the mean ± SEM.

The detection of major membrane lipids (PC, PE, PI, PS, PG and SM—check Figure 1 for meaning of abbreviations) and neutral lipids (Cer and TG) had good linear response against protein quantity and showed similar intensities in the analysis of both homogenates (Figure 1A). Overall, the behavior of summations for those lipid classes reflects that of individual species in each class (Supplementary Table S1). In the case of CL, the results did not show a good linear response in both extracts (Figure 1B), which indicates that this unexpected result is not a consequence of homogenization in Trizol. Although this seems to suggest that the extraction method is not quantitative for CL, when we normalized the intensity values of the lipid species with those of the internal standards (Cardiolipin Mix I from Avanti Polar Lipids) included before lipid extraction, both homogenates showed good linear response and similar slopes. This emphasizes the necessity of including lipid standards in extraction protocols not only because it allows for quantification, but also because it can correct procedure errors in minority lipids.

In the case of PCe and lysophospholipids, the slopes are substantially distinct between PBS and Trizol homogenates, although the $R^2$ of the linear regression is good in general. In

the case of PEe, not only were slopes substantially distinct, but in addition, Trizol samples did not show a coherent response against protein quantity.

We cannot exclude the possibility that during sample processing some PE and PEe may degrade and produce lysoPE. Arachidonic acid (20:4) is the most abundant fatty acid in the sn-2 position of the glycerol backbone of PE and PEe species; palmitic acid (16:0) and stearic acid (18:0) occupy the sn-1 position in more than 60% of PE. Small differences in susceptibility to hydrolysis during sample processing in PBS and Trizol (which is acidic) might account for the differences in LPE composition. In fact, plasmalogens (most of the PEe species in ESI-positive data; not shown) are more sensitive to acidic hydrolysis [11], which is avoided in the buffered PBS homogenates.

To verify if acidic hydrolysis was responsible for the aforementioned differences, we repeated the lipid extraction and lipidomic analysis of samples equivalent to 0.5 mg of protein. In this case, sample homogenization was carried out as previously (in PBS or Trizol), but also in Trizol with strict control of the time (≤2 min) between homogenization and the initiation of the lipid extraction procedure, and in some cases with extra buffering with PBS in the lipid extraction. Afterwards, lipid extraction, lipidomic analysis and data processing were performed as done previously.

Figure 2 shows proportions between ether-containing major phospholipids and their correspondent lysophospholipids. Trizol homogenates showed a dramatic decrease in PEe/LPE proportion when the time between homogenization and lipid extraction was over 5 min and no extra buffering was introduced. This is in accordance with results shown in Figure 1 and Figure S1. When time was decreased to ≤2 min PEe/LPE proportion was restored and extra buffering with more concentrated PBS improved the proportion in both lipids to values above those observed in the control.

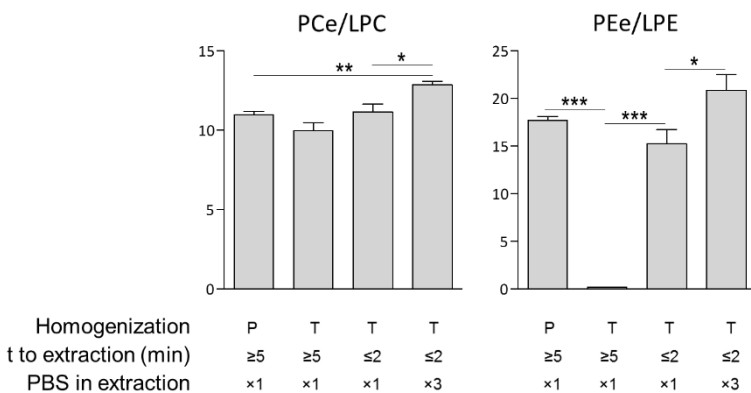

**Figure 2.** Proportions between major ether-containing phospholipids and their lyso-derivatives. Homogenization was performed in PBS (P) or Trizol (T), time between lipid extraction and homogenization was kept under 2 min or not and buffering of the lipid extraction mixture with PBS was variable (see Materials and Methods section). Data correspond to $n = 5$ and are shown as the mean ± SEM. Student's *t*-test: * $p < 0.05$; ** $p < 0.01$; *** $p < 0.001$.

In summary, this work shows that the kidney Tri-reagent homogenate is a suitable starting material for lipid extraction and lipidomic analysis by UHPLC-MS. An adequate procedure, in terms of time control and buffering of samples in homogenization and lipid extraction, leads to lipidomics results comparable to those obtained with samples homogenized in a usual buffer, such as PBS. Aiming to contribute an easy-to-follow guide that includes all mentioned steps, we provide a revised protocol for adequate sample processing and lipid extraction (Figure 3). This procedure does not hinder any of the steps of the RNA extraction protocol, as it only entails sharing the tissue sample for lipid and RNA extraction. Using the same starting material for both transcriptomic and lipidomic analyses will improve the alignment of results from both omics platforms. This will be useful, especially in the case of highly heterogeneous tissues, such as the kidney, because

even adjacent small fragments can have different phenotypes, including lipidome and transcriptome [3,4,12].

**Sample storage**

**1.** Freeze 20 – 100 mg of fresh tissue (preferably in liquid $N_2$) asap. Store at −80 °C.

**2.** Avoid any compound interfering with lipidomic analysis (such as detergents). Use RNase-free material.

**3.** Keep samples in solid $CO_2$ pellets until homogenization.

**Homogenization**

**4.** For quantitative lipidomic analysis, the exact tissue mass must be known. Quickly weigh the tubes containing the frozen samples. Transfer samples immediately to a suitable tube (13 mm diameter, RNase free) kept in $CO_2$ pellets. Weigh the empty sample tube and calculate sample mass by subtraction.

**5.** Add 0.6 – 1 mL of cold Tri-Reagent. Homogenize in ice bath with a Polytron coupled to a 12 mm dispersing aggregate, 4 – 6 intensity (or similar), for approx. 1 min or until complete homogenization. For RNA extraction, follow your usual storage protocol with a homogenate aliquot. The aliquot for lipid extraction should be used immediately (step 8).

**Lipid extraction**

Extraction procedure is based on Bligh and Dyer method [9]. Step 8 must be completed within 2 min after step 5. Use glass material washed with chromic mixture. Minimize the contact of solvents with plastic materials. Avoid unstable polymers such as polystyrene. Use high purity solvents (at least analysis grade), kept at room temperature.

**6.** In advance, prepare the starting solvent mixture in a suitable tube: 550 μL PBS (3× concentrated), 3 mL chloroform and 6 mL methanol per sample (see *diagram*).

**7.** Include internal standards (see *diagram*). We routinely use deuterated or non natural standards from Avanti Polar Lipids, 5 – 10 μl/sample of each:

> Splash I Lipidomix, Cardiolipin Mix I, Cer/Sph Mixture I, deuterated oleic acid, deuterated sulfogalactosyl ceramide and deuterated acylcarnitine

**8.** To initiate the extraction, transfer 50 μl of homogenate (equivalent to 1 – 5 mg of fresh tissue) to the starting solvent mixture.

**9.** Proceed to the first vortex mix and complete "*Extraction*" and "*Re-extraction*" (see *diagram*).

**Evaporation**

**10.** Dry extracts by centrifugal evaporation (Savant SpeedVac or similar) at 40 – 45 °C in the sample (approx. 1 h) (see *diagram*).

Extracts are ready to use.

**11.** If samples will be stored, dissolve the dry extract with 0.5 mL of chloroform:methanol 2:1 (*v:v*) and transfer it to adequate vials. Repeat step 10 to evaporate the solvent and store extracts at −80 °C in a $N_2$ atmosphere.

**Figure 3.** Optimized protocol for sample processing and lipid extraction starting from a tissue piece taken for RNA extraction with Tri-reagent [9].

**Supplementary Materials:** The following supporting information can be downloaded at: https://www.mdpi.com/article/10.3390/separations9100268/s1, Figure S1: Representative base peak intensity chromatograms of PBS- and Trizol-homogenized samples analyzed by UHPLC-MS. Figure S2: Distribution of lipid species among each lipid class; Table S1: Intensity of all lipid species identified in lipidomic analysis.

**Author Contributions:** Conceptualization, O.F. and Y.R.; Formal analysis, O.F., B.A.-G. and Y.R.; Funding acquisition, O.F. and Y.R.; Methodology, O.F. and Y.R.; Writing—original draft, O.F. and Y.R.; Writing—review and editing, O.F., B.A.-G. and Y.R. All authors have read and agreed to the published version of the manuscript.

**Funding:** This research was funded by the Basque Government (grants IT971-16, IT1476-22 and KK-2020/00069) and the Spanish Ministry of Science and Innovation (PID2021-124425OB-100).



**Institutional Review Board Statement:** The study was conducted in accordance with the Declaration of Helsinki, and approved by Ethics Committee on Animal Experimentation of the University of the Basque Country (protocol code: M20/2016/237 and date of approval: 2 December 2016).

**Informed Consent Statement:** Not applicable.

**Data Availability Statement:** Not applicable.

**Conflicts of Interest:** The authors declare no conflict of interest. The funders had no role in the design of the study; in the collection, analyses, or interpretation of data; in the writing of the manuscript; or in the decision to publish the results.

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
