# Peer review of "Tri-Reagent Homogenate Is a Suitable Starting Material for UHPLC-MS Lipidomic Analysis"

_separations, doi:10.3390/separations9100268_

Round 1
Reviewer 1 Report
The manuscript describes the novel work of using Tri reagent homogenate as a suitable starting material for lipid extraction and lipidomic analysis by UHPLC/MS. Protocol for sample processing and lipid extraction is very clear.
Part chemical and solutions is missing, also should be write the purity of the used standards and chemicals
After the manufacturer of chemicals reagents and equipment should be write the country in which the manufactured is located. also for Tri- reagent homogenate the data more in detail is missed.
For lipid extraction are used kidneys from 3 female Sprage-Dawley rats, is this a sufficient number for statistical data
All figures should be better quality and the text in the tables and figures should be corrected
The number of the used literature is very low so is needed to increase the number of the used literature
Reviewer 2 Report
The authors propose a new approach for lipid extraction combined with sample preparation for RNA isolation. This idea makes it possible to use the same material to obtain the two fractions simultaneously for further omics studies. The results from the new approach were compared with reference method (Bligh and Dyer lipid extraction method).
The manuscript can be recommended for publication after revision.
The following aspects of the presented study should be clarified, commented and/or revised:
1. 1. The terms Tri, Tri-reagent and Trizol appear to be used interchangeably, although this is not clearly indicated anywhere in the manuscript. Please unify through the text or clarify at the beginning.
2. 2. It is very difficult to follow the analytical protocol when the authors refer most of the analytical procedures to other publications (e.g. ref 2 and ref 5). It is recommended that they should be described at least briefly.
3. 3.The figure 3 should be mentioned and presented in the chapter 2.1 with updating the numbering of figures.
4. 4. It is also advantageous to see the samples preparation scheme summarized in the table taking into account the modifications applied (type of homogenate, concentration, extraction method, homogenization-extraction lag-time, buffering method, etc.).
5. 5. It is recommended to present the chromatogram(s) of the lipid extract (e.g. as TIC) obtained with both procedures for comparison.
6. 6. The use of an internal standard is mentioned only in Figure 3, while it should also be mentioned in Chapter 2.1.
7. 7. The authors indicated that the lipid extraction using Trizol may cause degradation of some lipid classes, and that time is detrimental. The solution proposed by the authors (shortening the time between steps 5 and 8) should be commented with respect to the practical considerations and the feasibility of colleting the two isolates (RNA and lipids) in a limited time and the quality of both resulting samples.
8. Supplementary table S1: Table cells have different colors, that were not explained previously.
8.
Reviewer 3 Report
This manuscript describes Tri-reagent homogenate is a suitable starting material for UHPLC-MS lipidomic analysis of kidney. The manuscript was well written. The manuscript may eventually be publishable, but requires minor revisions as indicated.
Minor scientific concerns
1.The novelty and advantages of this study should be illustrated in detail.
2. Is the Tri-reagent homogenate suitable for UHPLC-MS lipidomic analysis of other biological matrix?
3. The UHPLC-MS lipidomic analysis conditions should be provided.
